# Pre-clinical study of induced pluripotent stem cell-derived dopaminergic progenitor cells for Parkinson's disease

Daisuke Doi [1], Hiroaki Magotani[1,2], Tetsuhiro Kikuchi [1], Megumi Ikeda [1,3], Satoe Hiramatsu[1,3], Kenji Yoshida[1,3], Naoki Amano[4], Masaki Nomura[4], Masafumi Umekage[4], Asuka Morizane[1] & Jun Takahashi [1✉]

Induced pluripotent stem cell (iPSC)-derived dopaminergic (DA) neurons are an expected source for cell-based therapies for Parkinson's disease (PD). The regulatory criteria for the clinical application of these therapies, however, have not been established. Here we show the results of our pre-clinical study, in which we evaluate the safety and efficacy of dopaminergic progenitors (DAPs) derived from a clinical-grade human iPSC line. We confirm the characteristics of DAPs by in vitro analyses. We also verify that the DAP population include no residual undifferentiated iPSCs or early neural stem cells and have no genetic aberration in cancer-related genes. Furthermore, in vivo studies using immunodeficient mice reveal no tumorigenicity or toxicity of the cells. When the DAPs are transplanted into the striatum of 6-OHDA-lesioned rats, the animals show behavioral improvement. Based on these results, we started a clinical trial to treat PD patients in 2018.

[1] Department of Clinical Application, Center for iPS Cell Research and Application, Kyoto University, Kyoto, Japan. [2] Drug Safety Research Laboratories, Shin Nippon Biomedical Laboratories, Ltd, Kagoshima, Japan. [3] Regenerative and Cellular Medicine Kobe Center, Sumitomo Dainippon Pharma Co., Ltd, Kobe, Japan. [4] Facility for iPS Cell Therapy, Center for iPS Cell Research and Application, Kyoto University, Kyoto, Japan. ✉email: jbtaka@cira.kyoto-u.ac.jp

The discovery that the transplantation of fetal midbrain tissues can restore neurological symptoms of Parkinson's disease (PD) patients[1–4] has stimulated interest in cell therapies for the disease. Human pluripotent stem cells (PSCs), such as embryonic stem cells (ESCs) and induced PSCs (iPSCs), are sources for a stable supply of cells for the therapy[5]. Culture protocols that induce authentic midbrain dopaminergic (DA) neurons from human PSCs have already been established by several research groups, and the DA neurons have been proven safe and effective in rodent[6,7] and primate[8] PD models. These results suggest that PSC-derived DA neurons can contribute to the treatment of PD. However, the use of PSCs for cell-based therapies is such a new challenge that regulatory rules are not universally standardized. Indeed, each country has its own regulations and different criteria for clinical application. Therefore, to develop a global standard cell-based therapy for PD, it is important to share the results of pre-clinical studies and the outcomes of clinical trials[9].

Currently, two clinical trials using human ESCs are ongoing in Australia (NCT02452723) and China (NCT03119636), and their pre-clinical studies are reported[10,11]. We recently started a clinical trial (JMA-IIA00384, UMIN000033564) in Japan to treat PD patients by using iPSC-derived DA progenitors (DAPs). Because in these therapies the grafted cells survive and function as DA neurons for a long time (possibly until the patient dies), they must undergo more intensive risk management compared to other clinical trials using mesenchymal stem cells[12], in which the grafted cells function to provide neurotrophic support or immunomodulation but not as DA neurons, and do not survive long. Since 2015, we have continued discussion with the Japanese regulatory agency, Pharmaceuticals and Medical Devices Agency (PMDA), about details of the pre-clinical study. In this report, we show that the DAPs have no tumorigenicity or toxicity and that they improve the abnormal behavior of 6-hydroxydopamine (6-OHDA)-lesioned rats. Thus, we confirm the safety and efficacy of iPSC-derived DAPs for clinical trials.

## Results

**Manufacturing iPSC-derived DAPs under good manufacturing practice (GMP) conditions.** As a starting material, we used a human iPSC line (QHJI01s04) derived from the peripheral blood cells of a human leukocyte antigen (HLA)-homozygous healthy volunteer[13,14] and established for clinical application at the Center for iPS Cell Research and Application, Kyoto University, Japan. Then, we established a clonal master cell bank, MCB003, for our project. We have preserved the iPSCs in hundreds of frozen vials as one lot. To treat a patient, we thaw one of the vials, differentiate the cells into DAPs, and transplant them without freezing. In this pre-clinical study, we used multiple vials to confirm reproducibility of the results.

We modified our culture protocol to induce DAPs from human iPSCs[8,15] to meet the GMP-grade culture condition (Fig. 1a–d, Supplementary Table 1). Briefly, human iPSCs (MCB003) were seeded onto 6-well plates coated with laminin 511-E8 fragment and cultured for 12 days, and then CORIN+[16] cells were isolated by fluorescence-activated cell sorting (FACS). The sorted cells were cultured in 96-well plates as cell aggregates until day 30 and then used for the transplantation. In this culture condition, all reagents including an anti-CORIN antibody were at GMP grade. A clinical-grade cell sorter with a disposable fluid tube was equipped to mitigate the risk of cross-contamination.

**Characterization of iPSCs (MCB003) in vitro.** In the process of inducing DAPs, quality control check points were placed at the stage of MCB003, pre- and post-FACS sorted cells, and the final product (Fig. 1e). The quality standards of MCB003 include the morphology of the colonies, the expression of pluripotent markers such as TRA-1–60, TRA-2–49, and SSEA-4, and sterility. We confirmed that almost all cells expressed pluripotent cell markers (Supplementary Table 2).

**Characterization of the final product in vitro.** We conducted the DAP induction process in 25 randomly chosen vials from MCB003 (one time for each vial). Before sorting on day 12, the percentages of TRA-1–60+ and CORIN+ cells were 0.0% ($n =$ 24) and $31.4 \pm 12.7\%$ ($n = 25$), respectively. After sorting, the percentage of CORIN+ cells were $93.2 \pm 2.1\%$ ($n = 25$), confirming purity (Table 1, Supplementary Fig. 1a, b). The in-process testing for the manufacturing of DAPs is described in Supplementary Table 3.

The quality standards of the final product were determined to confirm that the cells were mainly composed of DAPs and included no undifferentiated cells. For this purpose, the expression of markers for DAPs (FOXA2 and TUJ1), iPSCs (OCT3/4 = POU5F1, LIN28, and TRA-2–49), and proliferating early neural progenitors (SOX1, PAX6, and KI67) were examined by flow cytometry and quantitative reverse transcription PCR (RT-qPCR). Flow cytometry on day 26 revealed that the percentage of DAPs (FOXA2+TUJ1+) was $92.3 \pm 4.0\%$ ($n = 25$; Table 1 and Supplementary Fig. 1c, d). Immunostaining also showed that most cells expressed DAP markers such as LMX1A and FOXA2 (Fig. 1f), and a smaller population expressed more mature DA markers such as NURR1 and TH (Fig. 1g).

Regarding undifferentiated cells, immunostaining showed that day-26 spheres did not contain iPSCs (OCT3/4 and NANOG; Fig. 1h) nor early neural progenitors (SOX1 and PAX6; Fig. 1i). Flow cytometry revealed that OCT3/4+TRA-2–49+ double-positive cells were not detected. The same flow cytometry analysis of DAP samples spiked with 0.1% iPSCs or SOX1+PAX6+ double-positive cells confirmed that the detection levels meet the test specification (Table 1, Supplementary Fig. 1e, f). In addition, RT-qPCR analysis showed that the expression levels of POU5F1 (OCT3/4) and LIN28 were $0.08 \pm 0.15\%$ and $0.14 \pm 0.13\%$ ($n = 25$ each) those of undifferentiated iPSCs, respectively (Table 1, Supplementary Fig. 1g, h).

To further confirm that there were no residual iPSCs, day-26 spheres were cultured in a condition suitable for iPSCs for 2 weeks. In this experiment, we mixed 0.001 to 1.0% of iPSCs with $2 \times 10^5$ dissociated cells from the day-26 spheres and found that even 0.001% was enough to form visible iPSC colonies. Importantly, day-26 spheres without mixture did not show any colony formation (Supplementary Fig. 1i). Taken together, these results indicated that the final product contained no or minimal (<0.001%) undifferentiated iPSCs.

N\ext, we examined if the DAPs could differentiate into functional DA neurons. To evaluate the production of dopamine, day-26 spheres were cultured until day 56 on L-ornithine/laminin/fibronectin-coated plates, and dopamine release in response to high KCl stimulation was measured by liquid chromatography with tandem mass spectrometry (LC-MS/MS) ($n = 5$, Fig. 1j). The amount of dopamine was $3.30 \pm 2.72 \, \text{pmol} \, \mu\text{g}^{-1}$ DNA in the cultured media, $0.47 \pm 0.45 \, \text{pmol} \, \mu\text{g}^{-1}$ DNA in low KCl, and $1.46 \pm 0.65 \, \text{pmol} \, \mu\text{g}^{-1}$ DNA in high KCl. Upon electrophysiological analyses, we found 67% of examined neurons showed spontaneous action potentials (Fig. 1k, l) and 89% of neurons showed induced action potentials ($n = 18$, Fig. 1m). The resting membrane potential was $-49 \pm 15 \, \text{mV}$.

**Genomic, epigenetic, and single-cell analyses.** The results above suggest that there is no tumorigenic component in the final

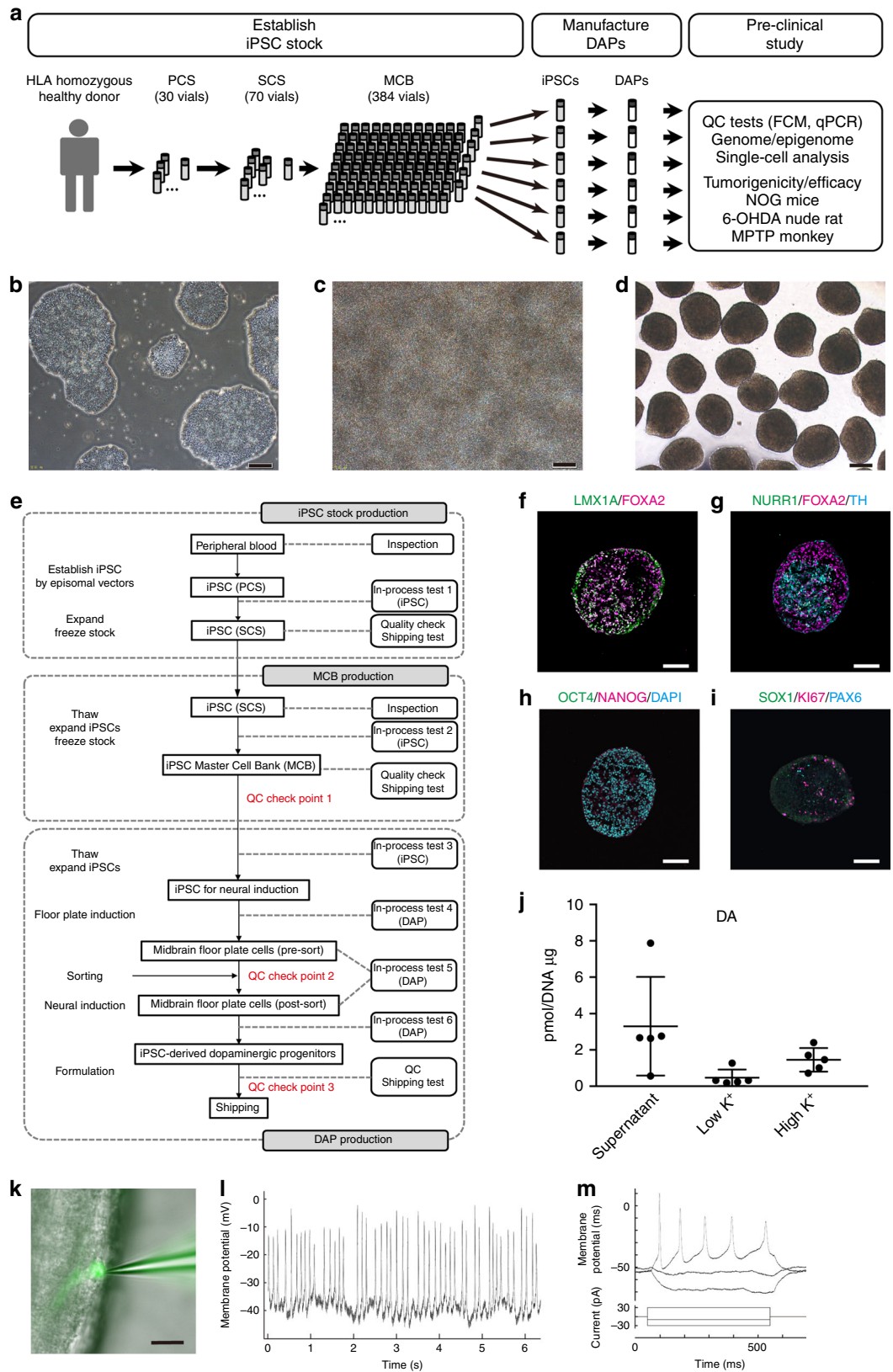

product. However, genomic and epigenetic changes in the cells may affect cell behavior after transplantation. To examine this possibility, genomic and epigenetic stability during the differentiation was examined, and the final products were subjected to tumorigenicity studies. We repeated DAP induction six times and compared the results of whole-genome sequencing (WGS) and whole-exome sequencing (WES) obtained from original peripheral blood cells, undifferentiated iPSCs, and differentiated cells on days 12 and 26.

We extensively investigated mutations in the 686 cancer-related genes listed in the Catalog of Somatic Mutations In Cancer (COSMIC) Census (COSMIC version 83). We also

**Fig. 1 Characterization of human iPSC-derived dopaminergic progenitors. a** An overview of the pre-clinical study. **b–d** Representative pictures of **b** iPSCs, **c** day-12 cultured cells before sorting, and **d** day-26 cultured aggregate spheres. Bars = 200 μm ($n = 25$ independent experiments). **e** Scheme of the cell production process in the GMP facility. **f–i** Immunofluorescence images of day-26 aggregate spheres sliced. Bars = 100 μm ($n = 25$ independent experiments). **j** The results of the LC-MS/MS analysis of the dopamine release by day-56 cultured cells under high potassium stimulation. Values are the mean ± SD ($n = 5$ independent experiments). **k** Representative image of whole-cell patch-clamp recordings. The patched cell was stained with Biocytin Alexa Fluor™ 488. Bar = 100 μm. **l** Representative current-clamp recording of spontaneous action potentials. **m** Representative current-clamp recording of action potentials induced by brief current pulses. **k, l** $n = 2$ independent experiments, the total number of cells = 18. PCS primary cell stock, SCS secondary cell stock, MCB master cell bank, QC quality control.

**Table 1 Characterization of the final product.**

| Test items | Test method | | Criteria | Results ($n = 25$) |
|---|---|---|---|---|
| Differentiation/undifferentiation markers | Flow cytometry (day 12) | TRA-1-60 CORIN | <1% (pre-sort) >10% (pre-sort) >90% (post-sort) | 0.0% ($n = 24$) 31.35 ± 12.72% 93.15 ± 2.127% |
| Morphology | Microscopic observation | | Sphere shaped, no foreign body | Complied |
| Viability | Flow cytometry: Cell count | | >90% | 97.48 ± 1.608% |
| DAP markers | Flow cytometry: FOXA2/TUJ1 | | >80% | 92.29 ± 3.964% |
| Undifferentiated markers | qPCR: *POU5F1* and *LIN28* | | <1% compared to undifferentiated cells | *POU5F1*: 0.08 ± 0.15% *LIN28*: 0.14 ± 0.13% |
| | Flow cytometry: OCT3/4/TRA-2-49/6E | | <0.1% | <0.1% |
| Early neural markers | Flow cytometry: SOX1/PAX6 | | <0.1% | <0.1% |
| Sterility | JP17 (membrane filter) | | Negative | Negative |
| Mycoplasma | JP17 (PCR) | | Negative | Negative |
| Endotoxin | JP17 (LAL turbidimetry test) | | ≤1 EU mL$^{-1}$ | Not analyzed |
| Residual plasmids | qPCR: Transfected genes | | Not detected | Not detected ($n = 6$) |
| CNV | SNP array Whole-genome sequencing | | No abnormalities compared to undifferentiated iPSCs in: | No abnormalities detected ($n = 6$) |
| SNV/Indel | Whole-genome sequencing Whole-exome sequencing | | (1) Shibata's gene list (Supplementary Table 4) (2) COSMIC census (3) HGMD database | |

LAL, limulus amebocyte lysate.

included 242 genes in Shibata's gene list (a cancer-related gene list made by the PMDA; Supplementary Table 4) and mutations annotated with the Human Gene Mutation Database (HGMD) PRO database (2016.4). No genomic mutations were detected by WGS (Table 2). WES revealed single-nucleotide variants (SNVs) in three genes, but this discovery was not consistently observed and thought to be a false positive after considering the results of the amplicon sequencing (Supplementary Tables 5, 6). In addition, there were no residual plasmids or copy number variations (CNVs) in all samples tested.

Regarding epigenetic modifications, we examined the methylation ratio at the transcriptional start site of 73 cancer-related genes. Our final products showed lower methylation levels than those of tissues with brain malignant tumor glioblastoma multiforme and similar levels with normal cells and fetal ventral mesencephalon (Fig. 2a and Supplementary Fig. 2).

To examine the variation of the gene expressions among the six independent preparations, we performed a single-cell RT-qPCR analysis. In each cell preparation, iPSCs on day 0, CORIN+ cells on day 12, and the final products on day 26 showed similar expression patterns in t-distributed stochastic neighbor embedding (tSNE) plots (Fig. 2b). In addition, there was no difference in the expression of genes related to DA differentiation (Fig. 2c, Supplementary Data 1). These results support the reproducibility of the cell manufacturing process.

**Tumorigenicity, toxicity, and biodistribution studies.** The same final products used for the genomic and epigenetic analyses were subjected to tumorigenicity studies. We repeated DAP induction

six times, and each DAP population was injected into the right striatum, which is the same target used in the clinical trial, of immunodeficient NOG (NOD.Cg-*Prkdc*$^{scid}$*Il2rg*$^{tm1Sug}$/ShiJic) mice. In this study, the toxicity and biodistribution of the cells were also examined. After discussion with the PMDA, we observed the animals as long as possible, namely for a life-long period. Initially, we grafted day-30 spheres ($2 \times 10^5$ cells per mouse, maximum dose) into 80 (males: 40, females: 40) and 50 mice (males: 25, females: 25) as the cell product-transplanted group and saline-injected control group (Supplementary Table 7), respectively, and observed the groups until the number of surviving male or female mice was 30 (transplantation group) or 20 (control group). At 52 weeks after transplantation (59–60 weeks old), the number of male mice in the control group was 19, at which point we stopped the observation. At this time, 28 males and 27 females survived in the transplantation group, and 19 males and 22 females survived in the control group. A survival curve of the animals showed no significant difference between groups (log-rank test, $p = 0.3208$, Fig. 3a). No changes were attributable to the transplantation in the mice's general conditions, body weight, food consumption, pharmacological conditions (modified Irwin test), ophthalmological findings, urinalysis, hematology and blood chemistry data, necropsy, or organ weight. Hematoxylin–eosin (H–E) staining of 5-μm-thick brain slices revealed no proliferative or malignant findings, and immunostaining for tyrosine hydroxylase showed that DA neurons had survived (Fig. 3b–e). In 23 (12 males and 11 females) of the 55 mice in the transplantation group, tubular cell clusters were found in the grafts and expressed pan-cytokeratin and

**Table 2 Summary of the genome analysis.**

| Preparation | Residual plasmid | SNV/Indel (Shibata's gene list/COSMIC Census-listed genes) | | CNV | Mutations in HGMD |
|---|---|---|---|---|---|
| | | WGS | Exome | | |
| CT1DAP-161004 | | | | | |
| Day 0 | None | 0 | 0 | 0 | None |
| Day 12 | NA | 0 | 0 | 0 | None |
| Day 26 | None | 0 | 0 | 0 | None |
| CT1DAP-161011 | | | | | |
| Day 0 | None | 0 | 0 | 0 | None |
| Day 12 | None | 0 | 0 | 0 | None |
| Day 26 | None | 0 | 0 | 0 | None |
| CT1DAP-161018 | | | | | |
| Day 0 | None | 0 | 0 | 0 | None |
| Day 12 | None | 0 | 0 | 0 | None |
| Day 26 | None | 0 | 0 | 0 | None |
| CT1DAP-161025 | | | | | |
| Day 0 | None | 0 | 0 | 0 | None |
| Day 12 | None | 0 | BARD1 | 0 | None |
| Day 26 | None | 0 | 0 | 0 | None |
| CT1DAP-161028 | | | | | |
| Day 0 | None | 0 | 0 | 0 | None |
| Day 12 | None | 0 | MUC4 | 0 | None |
| Day 26 | None | 0 | MUC4 | 0 | None |
| CT1DAP-161101 | | | | | |
| Day 0 | None | 0 | NCKIPSD | 0 | None |
| Day 12 | None | 0 | 0 | 0 | None |
| Day 26 | None | 0 | 0 | 0 | None |

NA, not analyzed.

transthyretin (TTR), suggesting that these cells have characteristics of choroid plexus epithelial cells (Supplementary Fig. 3a, b). The cell clusters were very small (<0.5 mm in diameter), not frequent (mean 1.4 clusters per graft), and not proliferating (no expression of KI67). In addition, they showed no effect on the survival or neurite extension of surrounding DA neurons. These results verified no tumorigenicity of the final products. Toxicity and biodistribution were also investigated using the same animals (see "Methods"), no abnormal findings were observed in the toxicity study, and grafted human cells (KU80+) were found only in the brain and not in systemic organs.

**Teratoma formation assay in subcutaneous space.** We further examined the existence of residual iPSCs in the final products by injecting day-30 spheres into the subcutaneous space of NOG mice with Matrigel, which is reported to be the most sensitive method to detect teratoma formation by residual undifferentiated iPSCs[17]. As a spike control, 0.001–10% of undifferentiated iPSCs (MCB003) were mixed into the day-30 spheres. As a positive control, another cell line of iPSCs (201B7, established by a retroviral vector with c-Myc[18]) and HeLa cells were used. When 600,000 HeLa cells or 201B7 cells were injected, subcutaneous tumors became detectable at around 10 weeks. In contrast, there was no tumor formation for 26 weeks in the DAP group, DAP with spiked MCB003 group, or even MCB003 group (Fig. 3f). Histological analysis revealed that KU80+ human cells survived in the DAP groups. On the other hand, no KU80+ cells were detected in the MCB group, suggesting that undifferentiated MCB003 cells cannot survive in the subcutaneous space (Fig. 3g–o). Similarly, MCB003 cells did not form any tumors in the mouse striatum, while 201B7 cells formed large teratomas

(Supplementary Fig. 3c). When MCB003 cells were injected into the testes of NOG mice, however, they formed typical teratomas, indicating that they were viable and pluripotent (Supplementary Fig. 3d). These results suggested that undifferentiated MCB003 cells do not remain in day-30 spheres, and that even if they did, the risk of teratoma formation is negligible.

**Efficacy study using 6-OHDA-lesioned rats and MPTP-treated monkeys.** To confirm the efficacy of clinical-grade iPSC (MCB003)-derived DAPs, day-30 spheres ($4 \times 10^5$ cells per rat) were transplanted into the striatum of 6-OHDA-lesioned nude rats ($n = 8$, vehicle control, $n = 6$), and methamphetamine-induced rotation was evaluated every 4 weeks. In the transplanted group, rotational asymmetry was reversed to normal levels by 16 weeks, while the control group showed no change in rotational behavior (Fig. 4a, two-way analysis of variance (ANOVA), Sidak's multiple comparison test, $p$ value: *** = 0.0002; ****<0.0001). Immunohistochemistry showed 2835 ± 2534 TH+FOXA2+ DA neurons survived and extended axons in the striatum (Fig. 4b–h).

For the clinical trial, we have developed a long-thin needle that is attachable to a stereotaxic frame. To examine the usability of the needle, MCB003-derived day-30 spheres (1.5–2.0 million cells per monkey) were transplanted into the left putamen of MPTP (1-methyl-4-phenyl-1,2,3,6-tetrahydropyridine)-treated monkeys ($n = 3$). Histological analysis 6 months after the transplantation revealed that the grafted DA neurons survived without any adverse effects, including hemorrhage or inflammation (Fig. 4i–n). The results of the in vivo pre-clinical studies are summarized in Supplementary Table 8.

### Discussion

For the clinical application of a cell-based therapy, a $2 \times 2$ matrix (Safety, Efficacy × Cellular and Non-cellular components; Supplementary Table 9) is recommended for quality control of the final product. In our case, the active cellular component is iPSC-derived DAPs. We confirmed that 92.3% of the total cells are FOXA2+TUJ1+ DAPs on culture day 26. We also confirmed that some of the survived human cells in rat were positive for GFAP, which suggests glial progenitors were included in the grafted cells with DAPs. The DAPs became mature DA neurons that produced dopamine and improved the abnormal behaviors of 6-OHDA-lesioned rats. Two previous clinical trials for PD using PSCs have transplanted ESC-derived neural progenitor cells (NCT02452723) and ESC-derived DA neurons (NCT03119636). These pre-clinical studies are summarized in Table 3.

There are three cellular components that should not be contained in the final product. The first is undifferentiated iPSCs, and we confirmed their absence by four methods in vitro. The second is early neural stem cells. Previously, we found that SOX1+ PAX6+ cells exist in the early stage of differentiation and that these cells form rosettes and proliferate in the brain[19]. The expression of CORIN, however, is never common with SOX1 or PAX6 expression. As a result, we can eliminate SOX1+PAX6+ cells by sorting CORIN+ cells. The third is transformed cells. The genomic instability of iPSCs and their derivatives may lead to tumorigenicity[20,21]. However, the need for genomic analysis is controversial, because genomic abnormalities do not always cause tumors. In addition, there is no standard gene list for which we should investigate. In this situation, we decided to investigate the genes listed in COSMIC Census and the cancer-related gene list made by the PMDA as well as familial disease-related genes listed in a HGMD PRO database. We performed WGS and WES of both iPSCs and the final products, but detected no aberrations in these genes. Epigenomic changes in iPSCs are another possible

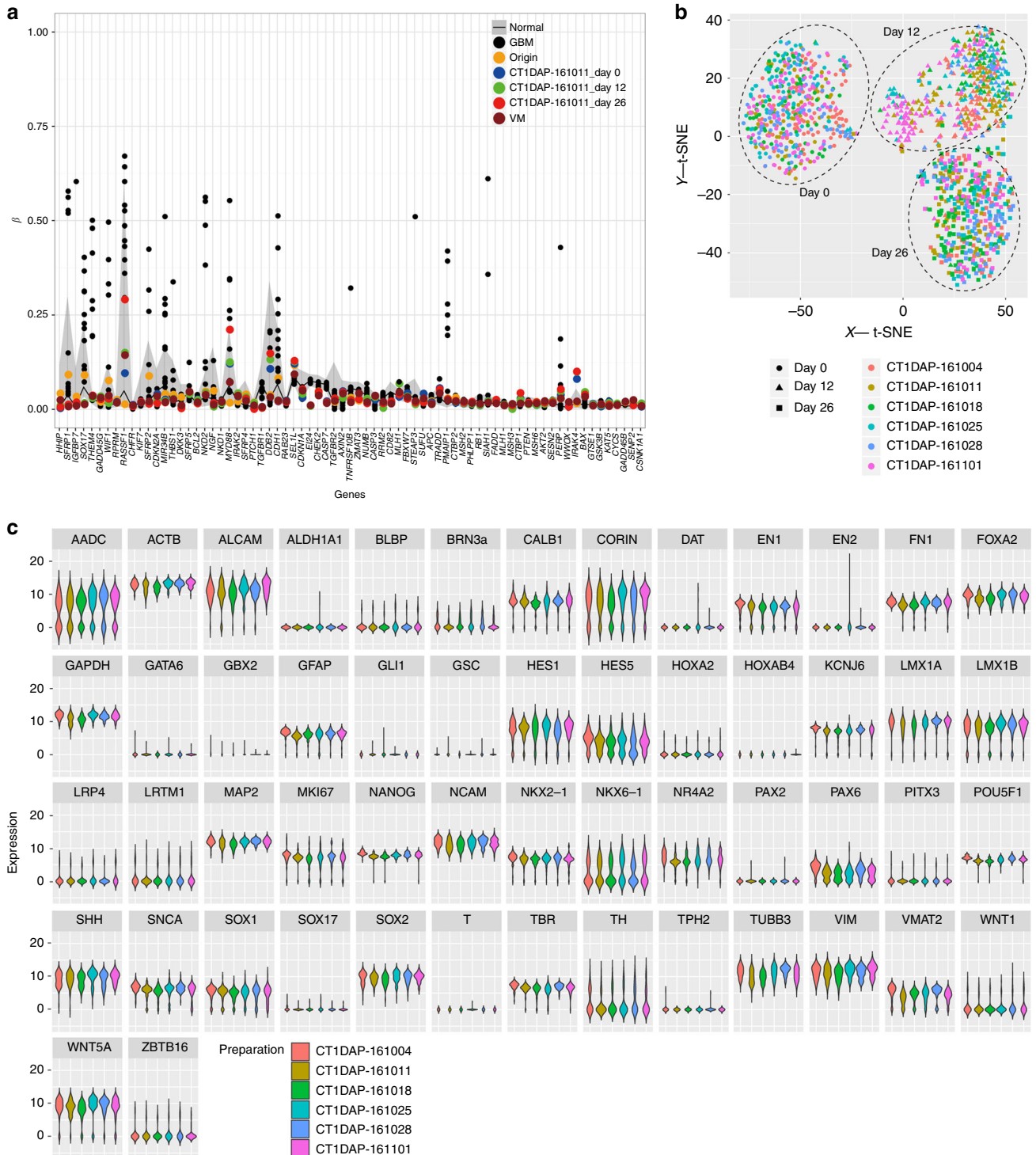

**Fig. 2 Genome/epigenome analysis and single-cell-based gene expression analysis. a** Representative results of the methylation array analysis of the DAPs (cell preparation: CT1DAP-161011) used in the tumorigenicity study. **b** A t-SNE plot of the single-cell data of the six lots tested in the tumorigenicity study. Each lot contains undifferentiated cells (day 0), intermediates (day 12), and the final product (day 26). **c** Violin plots of the expression pattern in DAPs from the six tested lots in the tumorigenicity study by single-cell RT-qPCR. GBM glioblastoma multiforme, VM ventral mesencephalon.

cause of tumorigenicity, because iPSCs are reported to have higher variability in DNA methylation[22]. From our results of a methylation array, iPSCs and the final products were not hyper-methylated compared with malignant brain tumor tissue and instead showed a methylation rate consistent with normal cells.

Tumorigenicity studies in vivo are also not standardized. In previous pre-clinical studies, the observation periods in nude rats and SCID (severe combined immunodeficiency) mice were 9 months (Table 3)[10,11,23]. In the case of spinal cord injury, ESC-derived oligodendrocyte progenitors (two million cells) were injected into the spinal cord of SCID mice, and the histological analysis was performed 12 months later[24]. In the present study, we used NOG mice, in which not only T cell and B cell activities but also NK cell activity is reduced, thus reducing the immune

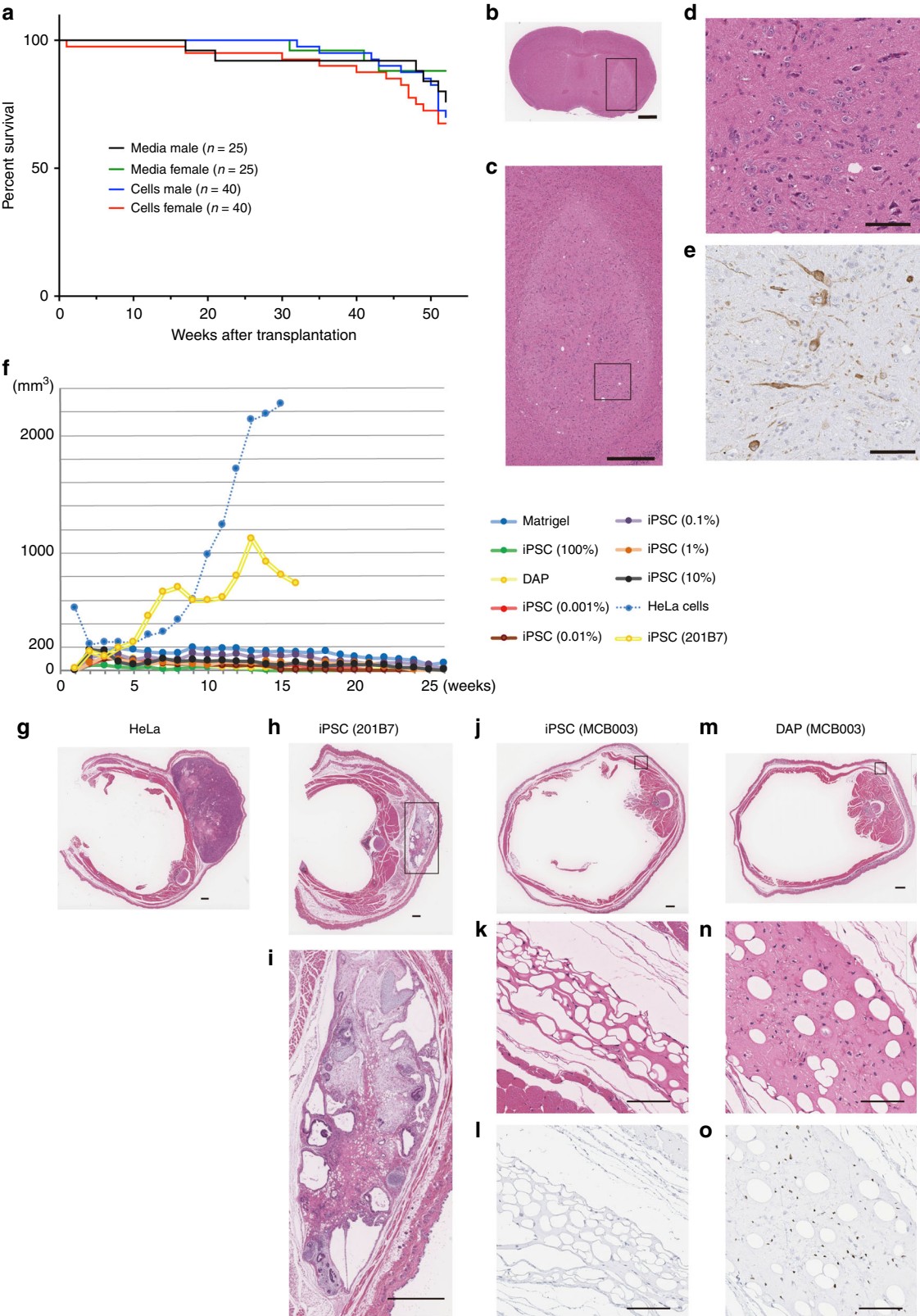

response[25]. The observation period was 52 weeks, which approximates the mouse lifespan. Because we found no proliferating cells or abnormal findings in the grafts, 52 weeks (or 12 months) seem to be long enough for tumorigenicity study. Of note, before starting the pre-clinical study, we consulted the PMDA about the study design. The PMDA requested we use NOG mice, which is the least immunogenic animal model, is sensitive to tumor development[17,26], and offers a long observation period. The PMDA also requested we evaluate at least 10 animals per group in the histological analysis. Major discussion points with the PMDA and our solutions are summarized in Supplementary Table 10.

**Fig. 3 Histological analysis of the tumorigenicity study. a** Survival curves of NOG mice tested in the tumorigenicity study. **b–e** Representative images of brain sections of the NOG mice used in the tumorigenicity study. **b–d** H–E staining (**c** is a magnification of **b**, and **d** a magnification of **c**) and **e** immunohistochemistry for tyrosine hydroxylase counter stained by hematoxylin. Bars in **b** = 1 mm, **c** = 500 μm, and **d**, **e** = 100 μm. Number of cell preparation = 6, and number of animals = 80 (transplantation) and 50 (control). **f** Time course of the subcutaneous mass volume of NOG mice used in the teratoma formation assay. Dots are mean values of each group. **g–o** Representative images of grafts in the subcutaneous space of NOG mice injected with HeLa cells (**g**, number of animals = 8), 201B7 iPSCs (**h**, **i**, number of animals = 6), MCB003 iPSCs (**j–l**, number of animals = 20), and MCB003 DAPs (**m–o**, number of animals = 21). **i** is a magnification of **h**, **k**, **l** are magnifications of **j**, and **n**, **o** are magnifications of **m**. **l**, **o** Staining by KU80. Bars in **g**, **h**, **j**, **m** = 1 mm, in **i**, **k**, **l**, **n**, **o** = 100 μm. **g–i** Number of cell preparation = 1 and **j–o** number of cell preparations = 7.

Regarding non-cellular components, we need to use clinical- or GMP-grade materials and reagents. We have made a GMP-grade anti-CORIN antibody and carefully confirmed that its remnant on cells was negligible. Regarding donor eligibility for iPSCs, a medical interview and blood exam for contamination were conducted. For other materials, γ-ray irradiation and viral elimination were applied to meet Japanese standards for biological raw materials. In addition, following guidelines of the International Council for Harmonization of Technical Requirements for Pharmaceuticals for Human Use (ICH) Q5A, we confirmed that the clinical-grade iPSCs (MCB003) were not contaminated with pathogenic viruses (Supplementary Table 11).

Unexpectedly, choroid plexus epithelia-like cells were observed in 23 of 55 mice with transplantation of the final product. In all cases, the cell clusters were microscopic (<0.5 mm), not frequent, not proliferating, and asymptomatic. Because bone morphogenetic protein (BMP) signaling is essential for developing choroid plexus tissue[27], we modified our differentiation protocol to extend the period of the BMP inhibition (Supplementary Fig. 4a). Accordingly, we successfully reduced the expression of TTR, a marker for the choroid plexus, in the final product, without any changes of other DA markers (Supplementary Fig. 4b, c). Transplantation to nude rat resulted in TTR+ cells appearing in two out of seven rats without the BMP inhibitor, but in no rats out of six rats with the BMP inhibitor (Supplementary Fig. 4d). We also confirmed that extension of the BMP inhibition can reduce the risk of inducing TTR+ epithelium-like cells in the brain.

In conclusion, the safety and efficacy of clinical-grade cell products were confirmed in this study. Based on these results, we received approval to start a clinical trial for PD from the PMDA and the institutional regulatory board of Kyoto University. The trial is ongoing.

## Methods

**Establishment of iPSC stock and manufacturing of DAPs**. This study was approved by the ethics committee of Kyoto University Hospital and Center for iPS cell Research and Application (CiRA), Kyoto University, Kyoto, Japan. After informed consent was obtained from an HLA-homozygous volunteer donor, an iPSC stock for clinical use was established by reprogramming the donor's peripheral blood with episomal plasmid vectors expressing *OCT3/4*, *SOX2*, *KLF4*, *L-MYC*, *LIN28*, *mp53DD*, and *EBNA1* at a cell processing center (Facility for iPS Cell Therapy, CiRA). The peripheral blood cells were isolated by using Ficoll-Paque PREMIUM (GE Healthcare), and $1.2 \times 10^7$ mononuclear cells were cultivated with StemFit AK03 without solution C (contains basic fibroblast growth factor) media (Ajinomoto) with 50 ng mL$^{-1}$ interleukin-6 (IL-6), 50 ng mL$^{-1}$ stem cell factor, 10 ng mL$^{-1}$ thrombopoietin, 20 ng mL$^{-1}$ Flt-3 ligand, 20 ng mL$^{-1}$ IL-3, and 10 ng mL$^{-1}$ granulocyte colony-stimulating factor (all WAKO) in four wells of a 24-well plate ($3 \times 10^6$ cells per well). After 7 days of cultivation, the vectors were induced in $5 \times 10^6$ dissociated cells by a Nucleofector 4D electroporation system (Lonza), and the cells were replated on laminin 511-E8 fragment (iMatrix, Nippi)-coated 6-well plates ($1.67 \times 10^5$ cells per well) in the same media as the mononuclear cells. One milliliter per well of StemFit AK03 media (Ajinomoto) was added 3, 5, and 7 days after the induction, and 9 days onward StemFit AK03 media were exchanged every 3 days. After 3 weeks of cultivation, single-cell-derived colonies became visible, and we picked up 15 of them manually. Each colony was dissociated with TrypLE Select CTS (Thermo Fisher), and all the cells were transferred to an iMatrix-coated 12-well plate and defined as passage 1 (P1). The P1 cells were passaged at $1.4 \times 10^3$ cells cm$^{-2}$ every 8–12 days, and 30 vials of a primary cell stock (PCS) were frozen at P4. Two out of fifteen PCS clones were identified as iPSCs by ESC-like morphology, the absence of

residual plasmids, confirmation of a normal karyotype, the expression of surface markers of undifferentiated iPSCs, and efficient neural differentiation. We selected the clones with the best efficiency for DA differentiation for the transplantation experiments. In order to confirm that the iPSCs were plasmid free by a longer culture, one vial of the PCS was thawed (P5) and expanded to 70 vials of secondary cell stock (SCS, P9). After thawing the SCS, we confirmed that there was no residual plasmids (tested at P9, P12, and P19), that they expressed undifferentiated markers (TRA-1–60, SSEA-4, and TRA-2–49; tested at P15), and had normal karyotypes (46, XY; tested at P12 and P13). One vial of the SCS was thawed and passaged two times, and then 384 frozen vials of MCB (P12) were stored. All freezing procedures were performed by using a programmed freezer. iPSCs were dispensed at $2 \times 10^5$ or $5 \times 10^5$ cells per tube in 200 or 500 μL of Stem Cell Banker GMP grade (TAKARA Bio, CB045), frozen to $-80\,°C$ at $-1\,°C$ min$^{-1}$, and stored in the gas phase of a liquid nitrogen tank.

Frozen MCB was differentiated after two passages of maintenance culture on iMatrix in StemFit AK03 medium was conducted. To manufacture the DAPs, the iPSCs were dissociated into single cells after 10-min incubation with 0.5× TrypLE select CTS (Thermo Fisher) and plated onto iMatrix-coated 6-well plates at a density of $5 \times 10^6$ cells well$^{-1}$ in differentiation media containing Glasgow's minimum essential medium (Thermo Fisher) supplemented with 8% γ-ray-irradiated knockout serum replacement (Thermo Fisher), 0.1 mM MEM (minimum essential medium) non-essential amino acids (Thermo Fisher), 1 mM sodium pyruvate (Sigma-Aldrich), and 0.1 mM 2-mercaptoethanol. A ROCK inhibitor, Y-27632 (final 10 μM), was added when the cells were dissociated, and the day of dissociation was defined as differentiation day 0. The media were changed daily, and 100 nM LDN193189 was added from days 0 to 12, 500 nM A83-01 was added from days 0 to 6, 100 ng mL$^{-1}$ fibroblast growth factor 8, and 2 μM purmorphamine were added from days 1 to 6, and 3 μM CHIR99021 was added from days 3 to 12. All materials and reagents used in this study are listed in Supplementary Table 12. On differentiation days 12 and 13, the cells were dissociated using 0.5× TrypLE select CTS, stained by phycoerythrin-conjugated anti-CORIN antibody (final concentration 100 ng mL$^{-1}$, Sumitomo Dainippon Pharma) for 20 min, and suspended with the sorting buffer containing phosphate-buffered saline (PBS)(−), 2% γ-ray-irradiated fetal bovine serum, 20 mM D-glucose, 80 μg mL$^{-1}$ gentamicin, and 10 μM Y-27632. Data analysis and cell sorting were performed with a FACSAria III or an Influx cell sorter (BD Biosciences). A 100-μm ceramic nozzle with a sheath pressure of 20 or 26 psi and an acquisition rate of ~5000 events s$^{-1}$ were used for the cell sorting. The positive gate was set so that <0.1% of cells were positive in unstained samples or samples stained by an isotype control antibody. Sorted CORIN+ cells were plated onto PrimeSurface® U-shaped 96-well plates (Sumitomo Bakelite) at a density of $2–3 \times 10^4$ cells well$^{-1}$ to make aggregate spheres in neural differentiation media containing Neurobasal medium with γ-ray-irradiated B27 supplement (without Vitamin A), 2 mM L-glutamine, 10 ng mL$^{-1}$ glial cell-derived neurotrophic factor, 200 mM ascorbic acid, 20 ng mL$^{-1}$ brain-derived neurotrophic factor, and 400 μM dibutyryl cAMP. Thirty micromolars of Y-27632 and 80 μg mL$^{-1}$ gentamicin were added at the initial plating after sorting, and the media were changed every 3 days until shipment or evaluation. The cell number of each aggregate sphere was calculated by the result of the flow cytometric analysis at day 26, and the number of spheres for animal transplantation was estimated by the cell number. At differentiation day 30, the aggregate spheres were collected in custom-made cell containers (JMS), washed with saline four times, and stored at 5 °C until shipment. A step-by-step protocol describing the DAP differentiation can be found at Protocol Exchange[28].

**Flow cytometry**. Flow cytometric analysis was performed using cells on differentiation days 12 and 26. On differentiation day 12, pre-sorted cells were stained by fluorescein isothiocyanate (FITC)-conjugated anti-TRA-1–60 antibody (1:5, BD #560380) for 20 min, followed by dead-cell staining by 7-aminoactinomycin D (1:100, BD #559925). On differentiation day 26, cultured spheres were dissociated to single cells by Neuron Dissociation Solutions S (FUJIFILM Wako Pure Chemical Corporation) and stained by the LIVE/DEAD® Fixable Dead Cell Stain Kit (Thermo Fisher) and FITC-conjugated anti-TRA-2–49 antibody (Merk-Millipore), followed by fixation by 4% paraformaldehyde for 30 min. After fixation, the cells were permeabilized by Perm/Wash buffer (BD Biosciences #554723) for 30 min, stained by antibodies for intracellular staining (Supplementary Table 13), and rinsed with Perm/Wash buffer. Stained samples were analyzed by Canto II or FACSAria III flow cytometer. The compensation setting of each fluorescence was

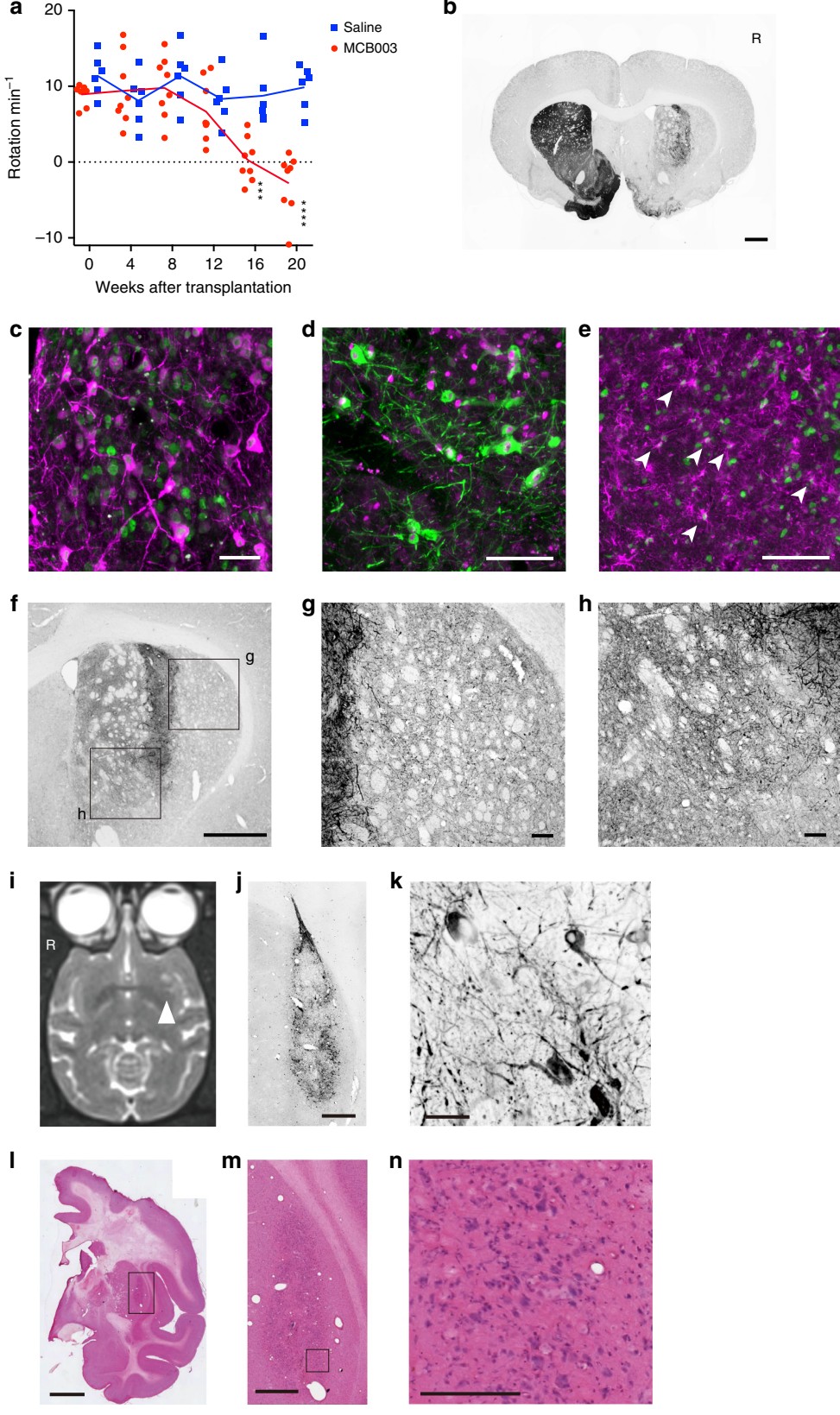

**Fig. 4 Results of the efficacy study. a** Rotational assays of methamphetamine-injected rats. Two-way ANOVA and Sidak's multiple comparison test, adjusted p value: *** = 0.0002 and ****<0.0001. **b–e** Representative images of the brain of a rat (number of animals = 8 for cell transplantation and 6 for saline injection) after transplantation and stained for **b** TH, **c** HNA (green) and TH (magenta), **d** TH (green) and FOXA2 (magenta), and **e** HNA (green) and GFAP (magenta). Bars in **b** = 1 mm, **c** = 50 μm, and **d**, **e** = 100 μm. R = right side of brain. **f–h** Representative images of the brain of a rat after transplantation and DAB stained for TH. **g**, **h** Magnified images of the boxes in **f**. Bars in **f** = 1 mm and **g**, **h** = 100 μm. **i–k** A magnetic resonance imaging (**i**) of the transplanted monkey and representative images (**j**, **k**) of the graft stained for TH. Arrowhead in **i** shows the grafts. Bars in **j** = 1 mm and **k** = 50 μm. **l–n** Representative H–E staining of the brain of a monkey after transplantation (**m** is a magnification of **l**, and **n** is a magnification of **m**). Bars in **l** = 5 mm, **m** = 1 mm, **n** = 200 μm. **b–n** Number of cell preparations = 2 and number of animals = 3.

adjusted by using the AbC Anti-Mouse Bead Kit and ArC Amine Reactive Compensation Kit (Thermo Fisher).

**Quantitative reverse transcription with polymerase chain reaction (qRT-PCR).** The total RNA fraction was extracted by NucleoSpin RNA XS (Takara Bio) and reverse transcribed by the SuperScript III First-Strand Synthesis System (Qiagen) or PrimeScript™ RT reagent Kit with gDNA Eraser (Takara Bio). Quantitative PCR was carried out with TaqMan™ Gene Expression Master Mix (Thermo Fisher) according to the manufacturer's instructions. The expression levels of the genes were normalized to *GAPDH* using the standard curve method. Probes and primers are as follows:

*GAPDH* probe, forward and reverse: Hs02758991_g1 (Thermo Fisher),
*POU5F1* probe: CGCATGGGGTTCGGCTTCCTGTCC,
*POU5F1* forward: GGTGGAGAGCAACTCCGATG,
*POU5F1* reverse: CAAATTGCTCGAGTTCTTTCTGC,
*LIN28* probe: CGCATGGGGTTCGGCTTCCTGTCC,
*LIN28* forward: CACGGTGCGGGCATCTG,
*LIN28* reverse: CCTTCCATGTGCAGCTTACTC.

**Immunostaining.** For in vitro studies, spheres were fixed with 4% paraformaldehyde and sliced at 10 μm thickness by using a cryostat (CM-1850, Leica Biosystems). Double- or triple-labeled immunostaining was performed after permeabilization, antigen retrieval if needed, and blocking with 0.1% Triton X-100 and 2% skim milk. The stained samples were visualized by using a fluorescence microscope (BZ-X710; Keyence) and a confocal laser microscope (Fluoview FV1000D; Olympus). The primary and secondary antibodies are listed in Supplementary Table 13.

**Single-cell analysis.** Single-cell complementary DNA (cDNA) preparation and gene expression analysis were performed following the manufacturer's protocol. A cell suspension of iPSCs (day 0), intermediate cells after sorting (days 12 and 13), or DAPs (day 26) at 300 cells μL⁻¹ were used. C1 Suspension Reagent (Fluidigm) was added to the cell suspension at a ratio of 40%. Six microliters of cell suspension mix were loaded on a C1 Single-Cell AutoPrep IFC microfluidic chip (Fluidigm) designed for cells 10–17 μm in size, and the chip was then processed on a Fluidigm C1 instrument using the "STA: Cell Load (1862×)" script. Live and single cells were confirmed under an inverted microscope. Then, the "STA: Preamp (1862×)" script was executed, which includes lysis, reverse transcription, and 18 cycles of PCR. When the run had finished, the amplified cDNA was harvested in a total of 28 μL C1 Harvesting Reagent. The amplified cDNA from single cells and DNA primers were then loaded onto a Biomark chip (Fluidigm) for expression profiling with a panel of qRT-PCR assays (Biomark HD, Fluidigm). The chip was then loaded onto Biomark HD to analyze gene expression levels. The expression data were processed and analyzed using the Singular Analysis Toolset version 3.6.2 (Fluidigm). The gene expression profiles of the analyzed cells were visualized using tSNE. A spectrum of gene expressions was visualized by violin plots. The sequences of the DNA primers are listed in Supplementary Table 14.

**Electrophysiological analysis.** To analyze the electrophysiological properties of cultured neurons, 42-day cultured spheres were attached on ʟ-ornithine/laminin/fibronectin-coated glass coverslips. After 14 days, whole-cell patch-clamp recordings were performed. Neurons on the surface of the spheres were chosen for the examination. The spheres were maintained in a physiological saline solution of the following composition: 125 mM NaCl, 2.5 mM KCl, 2 mM CaCl₂, 1 mM MgCl₂, 26 mM NaHCO₃, 1.25 mM NaH₂PO₄, and 17 mM glucose. Patch pipettes were made from borosilicate glass capillaries (1B150F-4; World Precision Instruments) and had a resistance of 3–4 MΩ when filled with an internal solution composed of 140 mM KCl, 10 mM HEPES, and 0.2 mM EGTA (pH 7.3). To visualize the patched cells, 100 μM Biocytin Alexa Fluor™ 488 was added to the internal solution. Recordings with a voltage clamp and current clamp were made with a patch-clamp amplifier (Axopatch 200B; Molecular Devices). The giga-seal resistances were in the range of 10–20 GΩ. The current signals from the amplifier were filtered at 1 kHz and stored and analyzed on a 64-bit computer (Cooler Master). All experiments were performed at room temperature.

**Dopamine measurement by LC-MS/MS.** Spheres on differentiation day 28 were replated onto ʟ-ornithine/laminin/fibronectin-coated 12-well plates with neural differentiation medium and cultured for 28 days. The medium was replaced 24 h before the measurements. On culture day 56, the cells were washed twice with low KCl solution (20 mM HEPES-NaOH, pH 7.4, 140 mM NaCl, 4.7 mM KCl, 2.5 mM CaCl₂, 1.2 mM MgSO₄, 1.2 mM KH₂PO₄, and 11 mM glucose) and incubated in low KCl solution for 15 min. The medium was subsequently replaced with 1 mL of high KCl solution (20 mM HEPES-NaOH, pH 7.4, 85 mM NaCl, 60 mM KCl, 2.5 mM CaCl₂, 1.2 mM MgSO₄, 1.2 mM KH2PO₄, and 11 mM glucose) for 15 min. The solution was then collected, and the concentration of dopamine was determined by LC-MS/MS (Sumika Chemical Analysis). Plated cells were harvested in PBS and sonicated. DNA concentration of the cell lysate was measured using the Quant-iT™ dsDNA Assay Kit (Thermo Fisher) and used for compensation of the DA concentration. Data were calculated from five independent experiments.

**Extended culture on iPSC maintenance condition.** Cells on culture day 26 or 30 were dissociated to single cells and replated onto iMatrix-coated 24-well plates at $2 \times 10^5$ cells per sample with StemFit AK03N iPSC maintenance media. Undifferentiated iPSCs were dissociated as well, and iPSCs were spiked into $2 \times 10^5$ DAPs at 0.001–1%. After 14 days, the cells were stained by alkaline phosphatase using the AP Staining Kit™ (System Biosciences), and the colonies were counted manually.

**Genomic/epigenomic analysis and testing residual plasmids.** WGS and WES libraries were prepared from 200 and 100 ng of genomic DNA as starting materials following the manufacturer's instructions. For WGS, libraries were made with the KAPA Hyper Prep Kit (Kapa Biosystems) without PCR. For WES, adapter-ligated libraries were prepared with the KAPA Hyper Prep Kit, and sequencing libraries were generated with the SeqCap EZ Human Exome Library v3.0 (Roche). Cluster generation was done with the HiSeq PE Cluster Kit v4-cBot (illumina) using illumina cBot. Sequencing was performed with the HiSeq SBS Kit v4 using HiSeq2500 and $2 \times 126$ PE mode. After generating FASTQ files by bcl2fastq v2.17.1.14 (illumina) and performing adapter trimming by cutadapt 1.10[29], FASTQ files were mapped to the reference human genome (hg19) by BWA MEM[30] (ver. 0.7.15) (for Genomon2, –T 0 option was used), and duplicated reads were removed by NovoSort (Novocraft) (ver. 1.03.09). To call SNV/Indels in test samples compared with donor cells, bam files were analyzed by Genomon[31] (ver. 1.0.1) with Fisher's exact test ($P < 0.001$) and Genomon2[32] (ver. 2.3.0) with EB call ($P$(Fisher) $< 0.1$ and $P$(EB call) $< 0.001$ for WGS, and $P$(Fisher) $< 0.1$ and $P$(EB call) $< 0.0001$ for WES), followed by adapting the custom filters. Mutations were annotated by ANNOVAR[33] (2016Feb01) and restricted on CDS and splicing regions. To further extract potentially pathogenic mutations, we excluded synonymous mutations and focused on mutations annotated with HGMD[34], COSMIC Cancer Gene Census[35] (ver. 83), and Shibata's gene list [http://www.pmda.go.jp/files/000152599.pdf]. WGS data were utilized for calling CNV using VarScan2[36] (2.4.2) in combination with Otsu's threshold method[37] and Delly[38] (0.7.3) by comparing test samples with donor cells, followed by curating the called CNVs manually. In addition to utilizing WGS data, CNVs were called using an SNP genotyping array. Genomic DNA (starting from 200 ng) was hybridized onto the HumanOmniExpress24 v1.2 DNA Analysis Kit (illumina), and intensities were scanned by iScan (illumina) following the manufacturer's protocol. After exporting a final report using GenomeStudio (V2011.1) (illumina), CNV analysis was performed using PennCNV[39] (1.0.3), Mosaic Alteration Detection-MAD[40] (1.0.1), and GWAS tools[41] (1.16.1), followed by comparing test samples with original cells and curating the called CNVs manually.

For the methylation analysis, the bisulfite conversion of 500 ng genomic DNA was performed using the EZ DNA Methylation Kit (Zymo Research), followed by methylation profiling using the Infinium MethylationEPIC BeadChip Kit (Illumina) according to the manufacturer's protocols. After exporting the methylation data by using GenomeStudio (V2011.1) and normalizing the beta values by using BMIQ[42], we calculated the beta values of 73 genomic blocks at the transcription start sites of cancer-related genes as described previously[43] and visualized them using R (https://www.R-project.org) with ggplot2[44]. As a reference, 53 methylation datasets of normal tissues (in GSE31848) and 11 methylation datasets of glioblastoma multiforme (downloaded from TCGA, Sentrix_ID: 9976861010) were also visualized.

Residual plasmids were tested by quantitative PCR with TaqMan™ Gene Expression Master Mix (Thermo Fisher). Quantitative PCR was done to amplify exogenous genes (*KLF4*, *SOX2*, *L-MYC*, *LIN28*, *Trp53DD*, *OCT3/4*, and *EBNA1*) in

**Table 3 Summary of pre-clinical studies using clinical-grade cells for cell therapies against Parkinson's disease.**

| Reference | Cellular product | Criteria | In vitro study | In vivo study | Number of animals | Cells/animal | Evaluation | Observation period |
|---|---|---|---|---|---|---|---|---|
| Garitaonandia et al. [10] | Human parthenogenetic stem cell-derived neural stem cells | NESTIN = 96.2% MSL1 = 93.5% SOX2 = 97.8% | Culture in ESC condition FCM QPCR | Nude rat | $N = 15$ $N = 4$ (control) $N = 80$ $N = 20$ (control) | $2.5 \times 10^6 - 7.7 \times 10^6$ $2.5 \times 10^5 - 7.7 \times 10^6$ | Acute toxicity Tumorigenicity/ biodistribution | 7 days 9 months |
| Wang et al. [11] | Human parthenogenetic ESC-derived DA neurons | TUJ1 = 99.8% TRA-1-60 = 0.1% SSEA-4 = 0.1% | Immunofluorescence FCM Electrophysiological analysis PCR | MPTP-treated monkey Nude mouse SCID mouse | $N = 3$ (FP), $N = 4$ (EB) $N = 3$ (control) Data not shown Data not shown | $2 \times 10^6$ Data not shown Data not shown | Tumorigenicity/ efficacy Tumorigenicity Teratoma formation | 9 months Data not shown 6 weeks |
| Present study | Human iPSC-derived dopaminergic progenitors | FOXA2/TUJ1 = 92.3% OCT4/TRA-2-49 < 0.1% SOX1/PAX6 < 0.1% | FCM QPCR Culture in iPSC condition Immunofluorescence Electrophysiological analysis DA measurement Genome/epigenome analysis Single-cell analysis | NOG mouse (brain) NOG mouse (subcutaneous) 6-OHDA-lesioned nude rat MPTP-treated monkey | $N = 80$ $N = 50$ (control) $N = 20$ $N = 60$ (iPSC-spiked) $N = 10$ (negative control) $N = 20$ (positive control) $N = 8$ $N = 6$ (control) $N = 3$ | $2 \times 10^5$ $6 \times 10^5$ $4 \times 10^5$ $1.5 \times 10^6 - 2.0 \times 10^6$ | Toxicity/ tumorigenicity/ biodistribution Teratoma formation Efficacy Efficacy/ tumorigenicity | 12 months 6 months 5 months 2–6 months |

FP, floor plate; EB, embryoid body.

the plasmid using iPSC genome DNA and real-time PCR Thermal Cyclers (StepOnePlus; Thermo Fisher). The DNA quantity was normalized to the copy number of endogenous *FBX15*. The DNA genomes of more than 25,000 cells were used as the template for the PCRs based on the copy number of endogenous *FBX15*. The probes and primers are as follows:

*KLF4* probe, forward and reverse: Hs00358836_m1 (Thermo Fisher),
*MYCL1* probe, forward and reverse: Hs00420495_m1 (Thermo Fisher),
*TP53* probe: CACAGTCGGATATCAGC,
*TP53* forward: GGGCAGCGGCTCTCTTGAG,
*TP53* reverse: CCAGGATGACTGCCATGGA,
*LIN28A* probe: CAAAAGGAGACAGGTGCTA,
*LIN28A* forward: CAAAAGGAAAGAGCATGCAGAA,
*LIN28A* reverse: CATGATGATCTAGACCTCCACAGTTG,
*SOX2* probe: ATCTCAAAATTGTCGCTCCT,
*SOX2* forward: GCCATTAACGGCACACTGC,
*SOX2* reverse: GAATTGTTCATGAGTGGACCTGG,
*OCT3/4* probe: TTCACCATGGCGGGACA,
*OCT3/4* forward: TGTCTCATCATTTTGGCAAAGAATT,
*OCT3/4* reverse: CGAGAAGGCGAAATCCGAA,
*EBNA1* probe: TGTCCGGAGACCCCA,
*EBNA1* forward: ATCAGGGCCAAGACATAGAGATG,
*EBNA1* reverse: GCCAATGCAACTTGGACGTT,
*FBX15* probe: GAATCAGACCTAACCACAGAGT,
*FBX15* forward: GCCAGGAGGTCTTCGCTGTA,
*FBX15* reverse: AATGCACGGCTAGGGTCAAA.

**Amplicon sequencing**. Primers for amplicon sequencing were designed with primer.py (Amelieff) and ApE (2.0.54) [https://jorgensen.biology.utah.edu/wayned/ape/]. For the multiplex PCR reaction, 10 ng of genomic DNA and 200 mM of primer mix (primers with KAPA HiFi HotStart ReadyMix (Roche) and 5% dimethyl sulfoxide) were mixed and PCR reaction was performed with the following condition: the initial denature was done at 95 °C for 5 min, followed by 35 cycles of denature at 98 °C for 20 s, annealing at 60 °C for 15 s, extension at 72 °C for 30 s, and final extension at 72 °C for 1 min. After purifying amplicon with 1.8x Agencourt AMPure XP Beads (Beckman Coulter), we made sequencing library with KAPA Hyper Prep Kit (Roche). Library was sequenced with MiSeq (Illumina) using MiSeq Reagents Kit v2 according to the manufacturer's instruction. The sequenced reads were aligned to the reference genome by BWA MEM (Ver. 0.7.15) and alternative allele frequencies at the targets were calculated by samtools (ver. 0.1.19)[45].

**Animal studies**. All animals were cared for and handled according to the Guidelines for Animal Experiments of Kyoto University and that of Shin Nippon Biomedical Laboratories (Kagoshima, Japan), and all animal experiments were certified by an ethics committee at Kyoto University and Shin Nippon Biomedical Laboratories.

**Tumorigenicity/toxicity/biodistribution study**. Seven–eight-week-old male and female NOG mice (CLEA Japan) were used for the transplantation. Cells were transplanted stereotactically, and the coordinates of the striatum were calculated with reference to the bregma: anterior (A), +0.5 mm; lateral (L), −2.0 mm; ventral (V), −3.0 mm; and tooth bar (TB), 0 mm. Two hundred thousand cells in 0.6–2.8 μL were injected through a 24 G needle (Hamilton 7105KH). Fifty-two weeks after the injection, the mice were euthanized with isoflurane and pentobarbital and perfused transcardially with PBS. The brains were immersion fixed by 10% formalin, paraffin embedded, and sliced with a microtome at 5 μm thickness throughout the graft. One of every 10 sections was used for H–E staining, and the other sections were used for immunohistochemisty. Other systemic organs (Supplementary Table 15) were also immersion fixed by 10% formalin and paraffin embedded and sliced with a microtome at 5 μm thickness. One of the sections was used for H–E staining, and the others were used for immunohistochemistry by anti-TH, anti-KU80, anti-KI67, anti-cytokeratin, and anti-TTR antibodies (Supplementary Table 13).

Toxicity and biodistribution analyses were combined with a tumorigenicity test of the brain injection in NOG mice. The test items evaluated are listed in Supplementary Table 15.

**Teratoma formation study (subcutaneous and testis injection)**. Five–nine-week-old male NOG mice were used. Two hundred fifty or 500 μL of dissociated single-cell solutions containing $6 \times 10^5$ cells mixed with ice-cold Matrigel (BD 356237) were injected by using a 22 G needle (Hamilton 750N) into the subcutaneous space of the upper back. The DAPs were spiked with 0.001–10% ($6 \times 10^0$–$6 \times 10^4$) undifferentiated iPSCs by the stepwise dilution method, and the cell concentration was confirmed to be within ±15% of the target concentration. The subcutaneous tumor size was measured by calipers, and the tumor volume was calculated via the following formula:

$$\text{Volume (mm}^3\text{)} = \text{Long diameter} \times (\text{short diameter})^2 / 2.$$

After 26 weeks of observation, the subcutaneous tissue was fixed by 10% formalin, and all tissues were paraffin embedded and sliced at 5 μm thickness for staining.

Animals in the positive control group (HeLa cells and iPSC line 201B7) were observed for 16 weeks or until the subcutaneous tumor volume was over 1500 mm³.

Ten microliters ($6 \times 10^5$ cells) of dissociated cells were injected into the right testis with a 22sG needle (Hamilton 702N). After 16 weeks of observation, the testes were extracted and fixed by FSA solution (formalin–sucrose–acetic acid). All fixed tissues were paraffin embedded and sliced at 5 μm thickness for staining.

**Efficacy study**. Transplantation of human iPSC-DAPs in 6-OHDA-lesioned rats: Adult (8–9-week-old) male nude rats (F344/NJcl-rnu/rnu, CLEA, Japan) were used for the lesioning. 6-OHDA injection was performed into the medial forebrain bundle, and only rats that showed more than six rotations per minute were used in the cell transplantation experiments. Cell transplantation was performed with the stereotactic injection of $4 \times 10^5$ cells (200,000 cells μL$^{-1}$) by using a 22 G needle (Hamilton 7105KH) into the right striatum (from the bregma: A, +1.0 mm; L, −3.0 mm; V, −5.0 and −4.0 mm; and TB, 0 mm).

Behavioural analysis: Methamphetamine-induced rotation behavior was evaluated before and every 4 weeks after the transplantation. The rotation was recorded for 90 min after the intraperitoneal injection of methamphetamine (2.5 mg kg$^{-1}$, Sumitomo Dainippon Pharma). The rotation number was counted by the RotoRat software version 2.01 (Med Associates, Inc., Vermont, USA).

Immunostaining: Twenty weeks after the transplantation, the rats were euthanized with isoflurane and pentobarbital and perfused with Ca$^{2+}$, Mg$^{2+}$-free PBS, and 4% paraformaldehyde. After post fixation for 1 day and immersion in 30% sucrose solution, the brains were sliced at 35 μm thickness for staining. Immunostaining was performed as in the in vitro studies. For 3,3′-diaminobenzidine (DAB) staining, sections were treated with 0.3% hydrogen peroxide in PBS with Tween 20 (PBST), incubated with primary antibodies, incubated with biotinylated antibodies against rabbit IgG (1:2000; Vector Laboratories), and then incubated with avidin peroxidase (1:4000; Vectastain ABC Elite Kit, Vector Laboratories). Signal detection was done by using DAB (Dojindo Laboratories) with nickel ammonium.

Transplantation of hiPSC-DAP in MPTP monkey by using custom-made needles: Adult male cynomolgus monkeys were supplied by Shin Nippon Biomedical Laboratories. Hemi-parkinsonian models were made by the intra-arterial injection of MPTP hydrochloride via the left carotid artery. A total of $1.5–2.0 \times 10^6$ cells per animal (four tracts, three deposits per tract) were transplanted into the left putamen of the monkeys with a customized surgical needle for the cell transplantation (TOP Corporation, Japan). After surgery, the monkeys were given antibiotics for 3 days and an intramuscular immunosuppressant (FK506, 0.05 mg kg$^{-1}$, Astellas) daily from one day before the transplantation to the day of sacrifice. The monkeys were subjected to magnetic resonance imaging scanning before and after the transplantation and finally sacrificed. Detailed descriptions of the monkey experiments are reported elsewhere[8].

**Statistical analysis**. Statistical analyses were performed by using the GraphPad Prism 7 software (GraphPad Software). Data from the survival curve of the transplanted animals were analyzed by the log-rank test (Fig. 3a), and the behavioral data were analyzed by a two-way ANOVA with Sidak's multiple comparisons test (Fig. 4a). $P$ values < 0.05 were considered significant.

**Reporting summary**. Further information on research design is available in the Nature Research Reporting Summary linked to this article.

## Data availability

Most of the data that support the findings of this study are available within the article and its Supplementary Information files or from the corresponding author upon reasonable request. Some of the data are not publicly available due to them containing information that could compromise donor privacy.

The datasets used in this study are as follows:

HGMD Pro 2016.4: https://digitalinsights.qiagen.com/products-overview/clinical-insights-portfolio/human-gene-mutation-database/

http://www.hgmd.cf.ac.uk/ac/index.php

COSMIC 83: https://cancer.sanger.ac.uk/cosmic

Shibata's gene list: http://www.pmda.go.jp/files/000152599.pdf

refGene, genomicSuperDups, snp131, esp6500siv2_all, 1000g2015aug_all were downloaded through ANNOVAR as humandb 20160720: https://doc-openbio.readthedocs.io/projects/annovar/en/latest/.

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

## Acknowledgements

We thank Sumitomo Dainippon Pharma Co., Ltd. for providing MCB003, and Dr. Y. Ono (KAN Research Institute) for kindly providing an anti-CORIN and anti-NURR1 antibody. We also thank Dr. Y. Maruyama (PMDA) for discussing the regulatory issues, Dr. P. Karagiannis for critically reading the manuscript, and Dr. E. Yamasaki, Dr. Y. Ozaki, Ms. Y. Ashida, Ms. Y. Tanikawa, Mr. Y. Fujita, Ms. Y. Katano, Ms. R. Takaichi, and Ms. Y. Ishii (CiRA, Kyoto University) for their technical assistance. This study was supported by a grant from the Network Program for Realization of Regenerative Medicine from the Japan Agency for Medical Research and Development (AMED).

## Author contributions

D.D. designed the study, collected and assembled the data, performed the data analysis and interpretation, and wrote the manuscript. H.M., T.K., M.I., S.H., and M.U. collected and/or assembled the data. K.Y., N.A., M.N., and A.M. performed the data analysis and interpretation. J.T. conceived and designed the study, assembled the data, carried out the data analysis and interpretation, wrote the manuscript, and made final approval of the manuscript.

## Competing interests

J.T. receives a grant for collaborative research by Sumitomo Dainippon Pharma Co., Ltd. M.I., S.H., and K.Y. are employees of Sumitomo Dainippon Pharma Co., Ltd. H.M. is an employee of Shin Nippon Biomedical Laboratories. The other authors declare no competing interests.
