## [Peer Review File · Nature Communications]

Reviewers' comments:

Reviewer #1 (Remarks to the Author):

This paper comes from one of the groups currently planning or carrying out clinical trials of human pluripotent stem cell-derived dopaminergic neurons to treat Parkinson's Disease, in this case using iPSC cells. As the authors point out, the requirements of different regulatory agents around the world differ, and it is of general interest to know the types and extent of preclinical data that have been obtained and required in different regulatory domains. In the present case, the authors described the preclinical data that they obtained to support their planned clinical trial in Japan. The report is valuable to those interested in understanding and deciding upon the types of data required before trials in man are undertaken. However, in the context of point made by the authors about differences between different regulatory regimes, it would have been interesting if they had included more in their Discussion about their discussions with the Japanese regulatory agency indicating what, if anything, was absolutely required, what may have been suggested, and if the authors carried out anything beyond what was required. Apart from that the manuscript does need some clarifications and I have the following comments:

1. In the Results the author briefly described the iPSC cell line used in the study, mentioning that they established a clonal master cell bank – and 'preserved the iPSCs in hundreds of frozen vials as one lot'. Given that this paper is giving details of the preclinical characterisation of the cells they were planning to use for their clinical study and, as I understand it, telling the readers about the preclinical data used to justify safety and efficacy, then I think they should provide more precision in the details of exactly how these particular iPSC cells were derived and the 'clonal master cell bank' established – how was the cloning done? how long were the cells grown before freezing? how were they frozen? How many vials are 'hundreds'? This information could be included in the Supplementary Data. Also, I presume that the regulators required data to indicate no retention of any of the reprogramming vectors? Is that the case and how was it achieved?
2. In a similar vein, when the authors say that they modified our protocol to induce DAPs from human iPSCs to meet the GMP-grade culture condition, I think they should give details of the modification – maybe the best would be to include in the supplementary data the SOP for their production of GMP grade DAPs.
3. I found that some of the Figures, notably Fig 1 b-c and Fig 3a as well as 3c-e and g-o, were too small and of too low magnification for me to make out anything useful. These figures should be at greater magnification and/or enlarged.
4. On page 3, under 'Characterisation of the final product in vitro' they authors say "We repeated the induction of DAPs from each randomly chosen vial of MCB003 25 times". Does this mean they did this from 25 random vials, testing each one once? Or did they induce DAPs 25 times from cells from just one vial.
5. On page 5, the details of the tumorigenicity studies need to be clarified. In particular, the authors write "Initially we grafted day-30 spheres into 80 (...) and 50 mice (....) as transplantation and control groups. I don't understand what is the control since it implies both groups were grafted with spheres! I can't read the Figure properly as it is too small but it looks as though the control is actually with medium injected rather than cells, but it is not clear in the text. The authors also conclude that there is no significant difference between the survival curves of the experiments and control mice – but from eye-balling the figure it looks as though the female experimental mice do have a slightly reduced survival; I cannot see which is the male experimental line.

Reviewer #2 (Remarks to the Author):

This study by Doi et al. contains all the safety and efficacy data that lead to the approval of their stem cell derived therapy for PD in Japan. We think it is extremely important to publish such data and to make it available for the scientific community. As such it is a highly valuable contribution to the stem cell therapy field. The paper is well structured and very well written, but the format for publication is a little different and we find it hard to review the paper as a regular scientific study as the data presented here have already been approved by the Japanese authorities.

We have therefore focused our evaluation on the presentation of the data, identifying where data could be presented clearer, and suggesting where additional text edits are recommended.

1) Characterization of final product in vitro:

- a. Add here the % of Corin and TRA1-60 positive cells before sorting.
- b. The limit of detection is stated to be 0.1% but this is a bit tricky to define and sensitivity varies between methods and instruments. Please refine this statement.
- c. The spike-in assay contains a group of 0.001% spiked in PSCs. How was the dilution made and how was it confirmed after mixing?

2) Tumorigenicity, toxicity and biodistribution

- a. DAPs from 6 separate differentiations were tested but it is unclear how many animals from each group were analysed.
- b. How was the maximum feasible dose determined?
- c. The teratoma assay is poorly described, please clarify.

3) Efficacy study

- a. Was the number of TH+/FOXA2 positive cells quantified?
- b. The extent of innervation in Figure 4 is not evident from the images provided.

4) Discussion

- a. It is stated that 92.5% of the cellular component is iPSC-derived DAPs. Is this in vitro at day 30 or after transplantation?
- b. It is stated that the rest are GABAergic neurons or midbrain astrocytes but these data are not presented in the result section.

5) Methods

- a. The cell differentiation protocol is not sufficiently detailed, It refers to two previous publications by the same group but there are slight differences in protocol used in the quoted Kikuchi et al and Doi et al. studies. We recommend to include all details of the protocol in this current manuscript.

6) Additional points

- The ability to reduce TTR-expressing choroid plexus like cells is not sufficiently described. Also, given that these cells only appeared in a less than 50% of the grafts means that a higher number of transplants need to be analysed in order to safely conclude that the presence of this cell type is reduced.
- The statement that the appearance of TTR-expressing choroid plexus like cells may be species specific should be nuanced. An equally plausible interpretation for why they do not appear in the primate grafts could be that only 3 animals were included in this group.

Reviewers' comments:

Reviewer #1 (Remarks to the Author):

This paper comes from one of the groups currently planning or carrying out clinical trials of human pluripotent stem cell-derived dopaminergic neurons to treat Parkinson's Disease, in this case using iPSC cells. As the authors point out, the requirements of different regulatory agents around the world differ, and it is of general interest to know the types and extent of preclinical data that have been obtained and required in different regulatory domains. In the present case, the authors described the preclinical data that they obtained to support their planned clinical trial in Japan. The report is valuable to those interested in understanding and deciding upon the types of data required before trials in man are undertaken. However, in the context of point made by the authors about differences between different regulatory regimes, it would have been interesting if they had included more in their Discussion about their discussions with the Japanese regulatory agency indicating what, if anything, was absolutely required, what may have been suggested, and if the authors carried out anything beyond what was required. Apart from that the manuscript does need some clarifications and I have the following comments:

According to the reviewer's comment, we added commentary about the regulatory requirements in the discussion section (line 263~) and a list in Supplemental table 8 (shown above). Detailed protocols for the iPSC and DAP generation were added in the methods section (line 294~).

1. In the Results the author briefly described the iPSC cell line used in the study, mentioning that they established a clonal master cell bank – and 'preserved the iPSCs in hundreds of frozen vials as one lot'. Given that this paper is giving details of the preclinical characterization of the cells they were planning to use for their clinical study and, as I understand it, telling the readers about the preclinical data used to justify safety and efficacy, then I think they should provide more precision in the details of exactly how these particular iPSC cells were derived and the 'clonal master cell bank' established – how was the cloning done? how long were the cells grown before freezing? how were they frozen? How many vials are 'hundreds'? This information could be included **in the Supplementary Data**. Also, I presume that the regulators required data to indicate no retention of any of the reprogramming vectors? Is that the case and how was it achieved?

According to the reviewer's comment, we added the detailed protocol of the iPSC stock generation in the methods section (line 294~). We chose one clone out of 15 picked

clones at P4, confirmed no residual plasmids at P9, P12, and P19, the expression of undifferentiated markers at P15, normal karyotype at P12 and P13, and the efficiency of neural differentiation at P9. Methods to detect residual plasmids were also written in the method section (line 504~).

2. In a similar vein, when the authors say that they modified our protocol to induce DAPs from human iPSCs to meet the GMP-grade culture condition, I think they should give details of the modification – maybe the best would be to include in the supplementary data the SOP for their production of GMP grade DAPs.

In the revised manuscript, we added details about the dopaminergic differentiation in the methods section (line 330~) and summarized the differences of the protocols between laboratory research and pre-clinical/clinical trials in Supplementary Table 1, which we also show below.

Supplementary Table 1. Modifications between laboratory-grade and GMP-grade cell production

	Laboratory Use (ref. 8)	Pre-clinical / Clinical trial
Sorting Buffer	PBS (-) or HBSS (-)	PBS (-)
	2% FBS	2% FBS, gamma-ray irradiated
	50 µg/mL Penicillin/Streptomycin	80 µg/mL Gentamycin
Cell Dissociation Enzyme	1×Accumax	0.5×TrypLE Select (with 0.5 mM EDTA/PBS (-))
Dead Cell Exclusion	7-AAD	None
Anti-CORIN Antibody	Gift from KAN Research Institute	Phycoerythrin-conjugated antibody (Sumitomo Dainippon Pharma)
Negative control	Stained with 2nd antibody	Unstained sample or stained with PE-isotype control
Differentiation media	KSR	Gamma-ray irradiated KSR
	Neurobasal/B27 w/o vitamin A	Neurobasal/B27 w/o vitamin A (gamma-ray irradiated B27)
	2 mM L-glutamate	2 mM Glutamax
	50 µg/mL Penicillin/Streptomycin	80 µg/mL Gentamycin
Replating plate	Lipidure-coated 96-well plate (U-shaped)	PrimeSurface 96U plate (Sumitomo Bakelite)

3. I found that some of the Figures, notably Fig 1 b-c and Fig 3a as well as 3c-e and g-o,

were too small and of too low magnification for me to make out anything useful. These figures should be at greater magnification and/or enlarged.

The figure sizes of question were unintentionally reduced during the initial submission process. They are enlarged in this session.

4. On page 3, under ‘Characterization of the final product in vitro’ they authors say “We repeated the induction of DAPs from each randomly chosen vial of MCB003 25 times”. Does this mean they did this from 25 random vials, testing each one once? Or did they induce DAPs 25 times from cells from just one vial.

We conducted the induction of DAPs in 25 random vials (one time for each vial). In all tested vials, the passage number of iPSCs before the induction was the same.

To avoid misunderstanding, we modified the expression in line 84 as follows.

“We conducted the DAP induction process in 25 randomly chosen vial from MCB003 (one time for each vial).”

5. On page 5, the details of the tumorigenicity studies need to be clarified. In particular, the authors write “Initially we grafted day-30 spheres into 80 (...) and 50 mice (...) as transplantation and control groups. I don’t understand what is the control since it implies both groups were grafted with spheres! I can’t read the Figure properly as it is too small but it looks as though the control is actually with medium injected rather than cells, but it is not clear in the text. The authors also conclude that there is no significant difference between the survival curves of the experiments and control mice – but from eye-balling the figure it looks as though the female experimental mice do have a slightly reduced survival; I cannot see which is the male experimental line.

We clarified the expression in line 162 of the revised manuscript as follows and enlarged figure 3a.

“Initially, we grafted day-30 spheres (2×10^5 cells/mouse, maximum dose) into 80 (male: 40, female: 40) and 50 mice (male: 25, female: 25) as the cell product-transplanted group and saline-injected control group (Supplementary Table 5), respectively, and observed the groups until the number of surviving male or female mice was 30 (transplantation group) or 20 (control group).”

It is true that in Figure 3a the female experimental mice appear to have slightly less survival, but we confirmed that the survival rate was not significantly different by the Log-rank test (shown in original manuscript line 165). We accordingly modified the expression in the revised text (line169).

“A survival curve of the animals showed no significant difference between ~~the transplantation and control~~ groups (Log-rank test, $p=0.3208$, Figure 3a).”

Reviewer #2 (Remarks to the Author):

This study by Doi et al. contains all the safety and efficacy data that lead to the approval of their stem cell derived therapy for PD in Japan. We think it is extremely important to publish such data and to make it available for the scientific community. As such it is a highly valuable contribution to the stem cell therapy field. The paper is well structured and very well written, but the format for publication is a little different and we find it hard to review the paper as a regular scientific study as the data presented here have already been approved by the Japanese authorities.

We have therefore focused our evaluation on the presentation of the data, identifying where data could be presented clearer, and suggesting where additional text edits are recommended.

1) Characterization of final product in vitro:

a. Add here the % of Corin and TRA1-60 positive cells before sorting.

According to reviewer’s comment, we added the percentages of Corin before sorting in the manuscript. TRA-1-60 was analyzed only before sorting. We revised the text to state this point in line 85.

“Before sorting on day 12, the percentages of TRA-1-60+ and CORIN+ cells were 0.0% (n=24) and $31.4\pm 12.7\%$ (n=25), respectively. After sorting the percentages of CORIN+ cells were $93.2\pm 2.1\%$ (n=25), confirming the purity of CORIN+ cells (Table 1, Supplementary Figure 1a).”

b. The limit of detection is stated to be 0.1% but this is a bit tricky to define and sensitivity varies between methods and instruments. Please refine this statement.

We validated the methods and instruments of the flow cytometry by using a spike-in assay. DAPs spiked with 0.1% iPSCs (OCT3/4 and TRA-2-49 positive) or early neural cells (SOX1/PAX6 positive) were measured by the same flow cytometry machine as used in the pre-clinical and clinical study, and OCT3/4/TRA-2-49 double-positive cells and SOX1/PAX6 double-positive cells could be detected in all 0.1%-spiked samples (n=3), which ensures our protocol meets test specifications. We modified the statement in the manuscript (line 101) and deleted comment about the lowest limit of detection and a reference in line 104 and Table 1.

“Flowcytometry revealed that OCT3/4+TRA-2-49+ double-positive cells were not detected. The same flowcytometry analysis of DAP samples spiked with 0.1% iPSCs or SOX1+PAX6+ double-positive cells confirmed the detection levels meet the test specification which is the lowest lower limit of detection (Table 1, Supplementary Figure 1c).”

c. The spike-in assay contains a group of 0.001% spiked in PSCs. How was the dilution made and how was it confirmed after mixing?

Samples of 200,000 undifferentiated iPSCs were diluted with a step-by-step dilution method to 0.001% (minimum 2 cells) and plated on a culture plate. After the cells were confirmed to exist in the culture plate by microscopic observation, the final products were immediately added on the culture plate. Although we did not confirm the precise number of spiked cells, we repeated the experiments and confirmed the reproducibility of the test. We added a photograph of the stained culture plates in Supplemental Figure 1f, as shown below.

Supplementary Figure 1f:

2) Tumorigenicity, toxicity and biodistribution

a. DAPs from 6 separate differentiations were tested but it is unclear how many animals from each group were analysed.

We added supplementary table 5 to show the number of animals per group per analysis.

Supplementary Table 5. Animal numbers used in tumorigenicity study

Differentiation number	Animal number			
	Control group		Sample group	
	Male	Female	Male	Female
CT1DAP-161004	5	5	2	2

CT1DAP-161011	5	5	12	12
CT1DAP-161018	5	5	7	7
CT1DAP-161025	5	5	8	8
CT1DAP-161028	5	5	6	6
CT1DAP-161101	0	0	5	5

b. How was the maximum feasible dose determined?

Our cells were transplanted as aggregate spheres. Before injection, the spheres were centrifuged briefly, and the supernatants were removed by aspiration, so that the cell concentration was maximized before transplantation (about 200,000 cells/ μ L). Injecting more spheres at 2 points/needle tract causes a backflow of aggregate spheres after removal of the needle. It is also technically hard to perform multiple injections precisely to mice striatum, and multiple injections raise the risk for bleeding and infection. As a conclusion, 200,000 cells per animal by one injection tract was maximum in our case.

c. The teratoma assay is poorly described, please clarify.

According to the reviewer's comment, we added details about the teratoma assay in the methods section (line 558~).

3) Efficacy study

a. Was the number of TH+/FOXA2 positive cells quantified?

In the original submission they were not quantified. We include the number of TH+FOXA2+ cells in the revised manuscript (line 214) and added immunohistochemistry data of TH and FOXA2 in Figure 4d, as shown below.

Figure 4d: Immunohistochemistry of grafted rat brain for TH (green) and FOXA2 (magenta). Bar = 100 μ m.

“Immunohistochemistry showed a $2,835 \pm 2,534$ substantial number of TH+FOXA2+ DA neurons survived and extended axons in the striatum (Figure 4b-hb, 4e).”

b. The extent of innervation in Figure 4 is not evident from the images provided.

In the revised manuscript, we provide images of DAB staining in grafted 6-OHDA rat in Figure 4f-h. The staining reveals fine fibers extended to the striatum.

Figure 4f-h. DAB staining of grafted rat brain for TH. Bars in f = 1 mm, g, h = 100 μ m.

4) Discussion

a. It is stated that 92.5% of the cellular component is iPSC-derived DAPs. Is this in vitro at day 30 or after transplantation?

We again examined FOXA2+TUJ1+ DAPs by ~~the QC test of~~ flow cytometry at culture day 26. We found the actual percentage is 92.3%. The revised manuscript has been corrected accordingly.

b. It is stated that the rest are GABAergic neurons or midbrain astrocytes but these data are not presented in the result section.

We added immunostaining data of grafted rats with glial marker GFAP and confirmed some GFAP positive cells were also positive for human nucleic antigen, although most of these cells were host-derived (Figure 4e; also shown below). On the other hand, we have no evidence that GABAergic neurons were in the graft, so we modified the expression in line 229 as below.

Figure 4e: Immunohistochemistry of grafted rat brain for HNA (green) and GFAP (magenta). Bar = 100 μ m.

“We also confirmed that some of the survived human cells in rat were positive for GFAP, which suggests glial progenitors were included in the grafted cells with DAPs.”

5) Methods

a. The cell differentiation protocol is not sufficiently detailed, It refers to two previous publications by the same group but there are slight differences in protocol used in the quoted Kikuchi et al and Doi et al. studies. We recommend to include all details of the protocol in this current manuscript.

According to the reviewer’s comment, we added details about the dopaminergic differentiation in the methods section (line 330~).

6) Additional points

- The ability to reduce TTR-expressing choroid plexus like cells is not sufficiently described. Also, given that these cells only appeared in a less than 50% of the grafts means that a higher number of transplants need to be analyzed in order to safely conclude that the presence of this cell type is reduced.

- The statement that the appearance of TTR-expressing choroid plexus like cells may be species specific should be nuanced. An equally plausible interpretation for why they do not appear in the primate grafts could be that only 3 animals were included in this group.

Additional experiments in rats found that 2 out of 7 rats expressed TTR+ cells in the absence of BMP inhibitor, while 0 out of 6 rats expressed TTR+ cells in the presence of BMP inhibitor (Supplementary Figure 4d). Although we analyzed only 3 monkeys, the total amount of transplanted cells was much higher in monkeys (5 million cells/monkey) than in mice (200,000 cells/mouse), so the total number of analyzed cells in 3 monkeys is equal to that in 75 mice.

However, we agree with the reviewer that we need to use more caution in our

conclusions. Therefore, we deleted the sentence commenting about species differences and added the following (line 285~).

~~The induction of such cells might be dependent on the host species or environment, because no epithelia-like cells were found in the monkeys despite the larger amounts of cells injected compared with mice.~~

“Transplantation to nude rat resulted that TTR+ cells were found in 2 out of 7 rats without BMP inhibitor, while no rats out of 6 rats with BMP inhibitor ((Supplementary figure 4d)). We also confirmed that the extension of BMP inhibition ~~avoided~~ **can reduce the risk of inducing** ~~the induction of~~ TTR+ epithelium-like cells in the brain.”

REVIEWERS' COMMENTS:

Reviewer #1 (Remarks to the Author):

The authors have addressed appropriately the concerns that I raised before

Reviewer #2 (Remarks to the Author):

We are fully satisfied with the authors revision and recommend the paper for publication in its present form.